# Requiem for Rimonabant: Therapeutic Potential for Cannabinoid CB₁ Receptor Antagonists after the Fall

**Taryn Bosquez-Berger** [1,2,3], **Gergő Szanda** [1,2,4] **and Alex Straiker** [1,2,3,*]

1. The Gill Center for Biomolecular Science, Indiana University, Bloomington, IN 47405, USA; tbosquez@iu.edu (T.B.-B.); gszanda@iu.edu (G.S.)
2. Department of Psychological and Brain Sciences, Indiana University, Bloomington, IN 47405, USA
3. Program in Neuroscience, Indiana University, Bloomington, IN 47405, USA
4. Department of Physiology, Semmelweis University Medical School, 1094 Budapest, Hungary
*  Correspondence: straiker@indiana.edu; Tel.: +1-206-850-2400

**Abstract:** The endocannabinoid system is found throughout the CNS and the body where it impacts many important physiological processes. Expectations were high that targeting cannabinoid receptors would prove therapeutically beneficial; pharmaceutical companies quickly seized on the appetitive and metabolic effects of cannabinoids to develop a drug for the treatment of weight loss. Alas, the experience with first-in-class cannabinoid type-1 receptor (CB₁R) antagonist rimonabant is a now-classic cautionary tale of the perils of drug development and the outcome of rimonabant's fall from grace dealt a blow to those pursuing therapies involving CB₁R antagonists. And this most commercially compelling application of rimonabant has now been partially eclipsed by drugs with different mechanisms of action and greater effect. Still, blocking CB₁ receptors causes intriguing metabolic effects, some of which appear to occur outside the CNS. Moreover, recent years have seen a startling change in the legal status of cannabis, accompanied by a popular embrace of 'all things cannabis'. These changes combined with new pharmacological strategies and diligent medicinal chemistry may yet see the field to some measure of fulfillment of its early promise. Here, we review the story of rimonabant and some of the therapeutic niches and strategies that still hold promise after the fall.

**Keywords:** rimonabant; accomplia; CB₁; cannabinoids; CB₁ antagonists; pharmacotherapy



## 1. The Endocannabinoid System

### 1.1. Receptors and Ligands

Cannabinoids have been in use for thousands of years [1,2], but systematic inquiry into how cannabinoids work in the body began only in the 1940s when chemists isolated chemical constituents of cannabis such as cannabidiol (CBD) and tetrahydrocannabinol (THC) [3] (Figure 1). Ultimately, more than a hundred chemically related phytocannabinoids were identified, but the question of how cannabinoids act in the body remained a mystery for decades. Cannabinoid research saw a flowering in the 1970s, with early indications that cannabinoids might be helpful as therapeutics for some specific ailments. Synthetic THC found use promoting appetite in AIDS patients and combatting nausea and vomiting in patients undergoing chemotherapy [4–6]. But, by the mid-1980s, this research effort had dissipated. The question of how cannabinoids act in the body remained unanswered until the identification of cannabinoid receptors. These receptors, dubbed CB₁ [7] and CB₂ [8], are part of a large family of proteins known as G protein-coupled receptors (GPCRs) that includes targets for opiates, dopamine, serotonin, acetylcholine and many more receptors involved in neuronal and non-neuronal signaling. Most medicines target GPCRs.

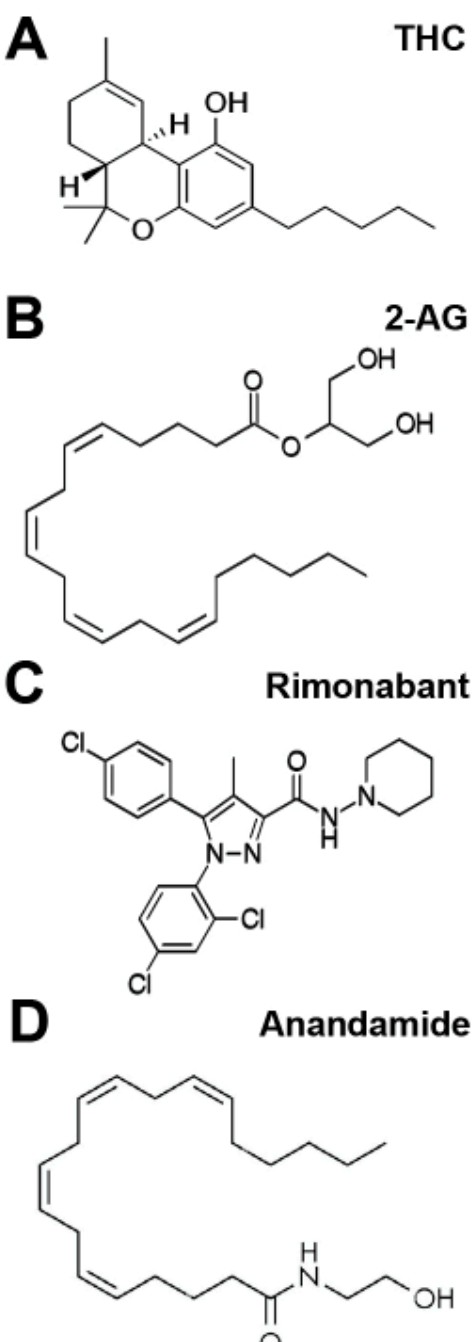

**Figure 1.** Structure of THC, 2-arachidonoyl glycerol (2-AG), rimonabant and anandamide.

As a consequence of the discovery of the cannabinoid receptors, we now know that cannabinoids act by plugging into an endogenous cannabinoid signaling system, similar to the way opium acts via opioid receptors in the body. Within a few years, two endogenous ligands were identified. These endocannabinoids are structurally unrelated to THC (Figure 1), consisting of arachidonic acid with distinct headgroups: 2-arachidonoyl glycerol (2-AG, [9]) and arachidonoylethanolamide (AEA [10]), though AEA is also commonly referred to as anandamide, from the Sanskrit for bliss.

*1.2. Endocannabinoid Metabolism*

Because endocannabinoids are membrane-preferring lipids, and in contrast to many neuronal messengers, they are not released at the synapse by vesicles, instead, they are produced enzymatically, 'on demand'. Enzymes such as diacylglycerol lipases [11] or

N-acyl phosphatidylethanolamine (NAPE)-phospholipase D [12,13] are activated to rapidly synthesize the endocannabinoids AEA or 2-AG, respectively. After endocannabinoids have activated their target receptors, they are then inactivated metabolically, modified or broken into their constituent parts that are recycled for other purposes. The number of enzymes that have been implicated in cannabinoid metabolism is not small, but the main roles have been assigned to monoacylglycerol lipase (MAGL) for 2-AG [14], and fatty acid amide hydrolase (FAAH) for AEA [15,16]. Endocannabinoid metabolizing enzymes typically act not just on arachidonoyl acid-based lipids but also the shorter chain oleoyls, palmitoyls and others. This has several biological consequences as some of the products of eCB synthetizing enzymes are ligands for other receptors as in 2-oleoylglycerol (2-OG) and GPR119 [17]. As a result, there is much current debate over what constitutes an endocannabinoid and a cannabinoid receptor. To complicate matters, AEA and 2-AG can both activate the TRPV1 receptor, best known as the receptor that is stimulated by chili peppers and heat [18]. Anandamide remains the strongest candidate endogenous ligand for this receptor. Consequently, the cannabinoid signaling system may encompass six or more receptors, at least as many endogenous ligands, and a stable of enzymes to produce and break them down. Moreover, endocannabinoids and endocannabinoid-derived arachidonic acid are substrates of cyclooxygenases and may thus serve as precursors for prostamides and prostanoids, that are active compounds with pleiotropic biological effects [19].

### *1.3. CB$_1$ Receptor—Localization and Function*

Since the discovery of endocannabinoids and cannabinoid receptors, the most attention has been paid to the canonical cannabinoid receptors, particularly CB$_1$R. Soon after the receptor was first described, researchers mapped out its distribution, finding that it is widely expressed in the brain (Figure 2; [20]) and throughout the body [21]. Indeed, there are few neuronal systems that do not express CB$_1$ receptors. And in contrast with most ligands for GPCRs, the lipophilic cannabinoids readily cross into the CNS. In principle therefore, cannabinoids represent a 'target-rich' therapeutic opportunity. The risk is that each site also represents a potential off-target effect. A life-saving treatment in the cerebellum might come with a perilous side-effect in the hippocampus, a subject to which we shall return.

Once it became clear where CB$_1$ receptors were expressed, the question became 'what are they doing there?'. One clue came from anatomical studies: CB$_1$ receptors tended to reside presynaptically, near the release site for neurotransmitters [22]. GPCRs act by converting an extracellular signal into an intracellular signal, often by initiating a signaling cascade that rapidly amplifies the signal throughout the cell. The kind of signaling they initiate and, ultimately, the biological consequence are defined by the G proteins to which they couple, and it was soon learned that CB$_1$ receptors primarily couple to G$_{i/o}$ G proteins, inhibiting calcium channels and adenylyl cyclase formation of cyclic AMP and activating the Raf/Ras/MEK signaling cascade (Figure 3), though other signaling pathways such as arrestin signaling also likely contribute to their function. In neurons, CB$_1$ activation is predominantly inhibitory in nature, reducing the amount of neurotransmitters released from neurons. Because endocannabinoids are typically produced post-synaptically, this means that the direction of effect is retrograde, from post-synapse to pre-synapse. This contrasts with classical neurotransmitters such as acetylcholine, glutamate and GABA. This means that, by and large, the neuronal role of CB$_1$R is to serve as a feedback inhibitor (reviewed in [23]). Because CB$_1$ receptors can inhibit either excitatory or inhibitory neurotransmitter release, the net consequence of CB$_1$ activation depends on the circuit; e.g., inhibiting an inhibitory circuit can result in a net excitation. In addition, new roles continue to be found for CB$_1$Rs in neurons and elsewhere (e.g., mitochondrial [24] or somatodendritic [25] CB$_1$Rs).

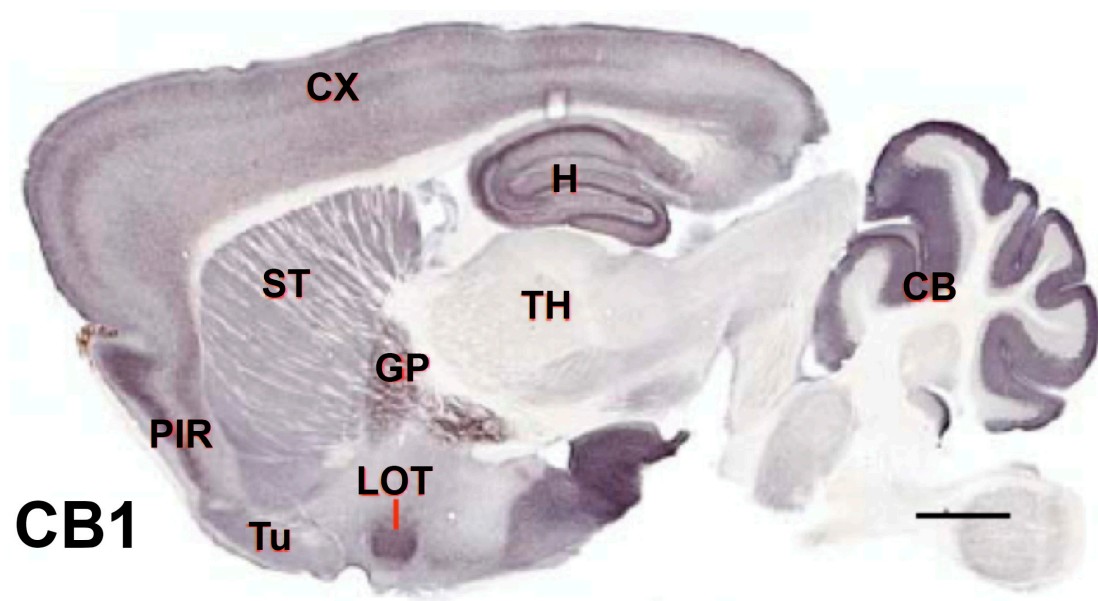

**Figure 2.** $CB_1R$ is widely distributed in the brain. With the notable exception of the thalamus (TH), the cannabinoid receptor type 1 ($CB_1$) is seen in most regions of the mouse brain including cortex (CX), hippocampus (H), striatum (ST), and cerebellum (CB). Other brain regions shown: piriform cortex (PIR), olfactory tubercle (Tu), lateral olfactory tract (LOT), and globus pallidus (GP). Source: Huei-Ying Chen.

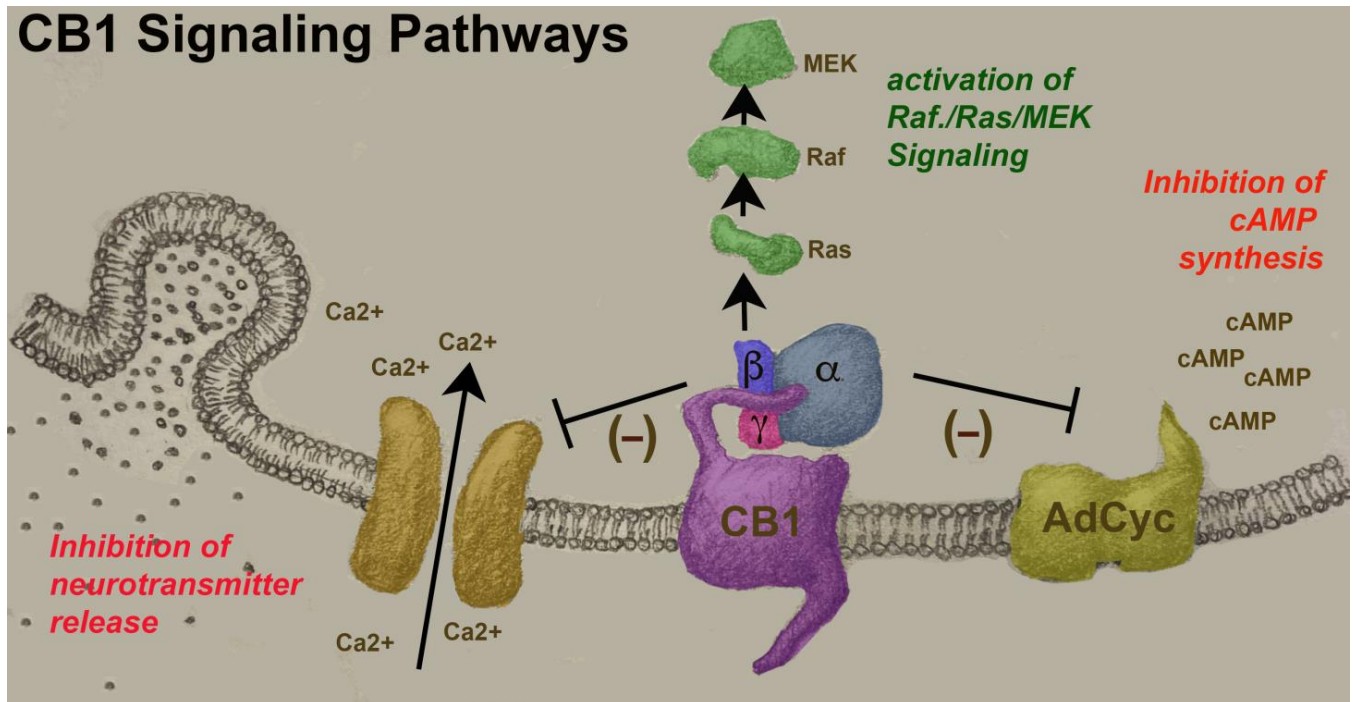

**Figure 3.** Canonical $CB_1$ receptor signaling pathways. In neurons, the cannabinoid receptor type 1 ($CB_1R$) acts via G proteins $\alpha$ (alpha), $\beta$ (beta) and $\gamma$ (gamma), to inhibit calcium ($Ca^{2+}$) channels (and consequently neurotransmitter release) and also inhibits adenylyl cyclase (AdCyc), thus reducing production of the intracellular messenger cyclic adenosine monophosphate (cAMP). The $CB_1$ receptor also activates the rat sarcoma virus (Ras)–rapidly accelerating fibrosarcoma (Raf)–mitogen activated protein kinase kinase (MEK) (commonly denoted as Raf-Ras-MEK) signaling pathway. Additional non-canonical pathways such as arrestin signaling also likely contribute to $CB_1R$ effects.

## 2. Targeting CB$_1$ Receptors with Pharmacological Tools

Once cannabinoid receptors were identified, researchers set about developing pharmacological tools such as synthetic receptor agonists and antagonists as well as blockers for metabolic enzymes. With these tools in hand, the list of potential therapeutic applications for cannabinoids grew rapidly. Beyond appetite and nausea, cannabinoids were investigated for roles in pain, inflammation, anxiety, addiction, neurodegenerative disorders, glaucoma, and more. The cannabinoid regulation of pain has been one of the most promising areas of research, with several drugs in clinical trials targeting enzymes such as FAAH. The reasoning is that by preventing the breakdown of anandamide, one could harness the *endogenous* cannabinoid signaling system. Though broadly encouraging, the research suffered a setback when a proposed FAAH blocker BIA-10-2474 resulted in a fatality in one subject and serious neurological complications in others in Stage I clinical trials. However, this appears to have been a consequence of the off-target effects of the drug rather than blocking FAAH [26,27]. A cannabinoid role in epilepsy was also explored, partly because CB$_1$ receptor knockout mice were more prone to seizures [28]. But the challenge of using an agonist that has complex effects on both excitatory and inhibitory synapses is that the net effect is difficult to predict. But intriguingly, the phytocannabinoid CBD was approved in 2018 as a treatment for a form of childhood epilepsy [29] and shows promise for other seizure disorders. The mechanism of action is still unclear but will be discussed below.

## 3. Rise and Fall of Rimonabant

### 3.1. The Prelude

One major dilemma for the development of cannabinoid therapeutics—the elephant in the room—is the psychoactivity of cannabinoids. Though the legal landscape is changing, back in the 1990s, at the outset of receptor-based cannabinoid pharmacology, the possibility that patients might experience cannabis-associated intoxication as a side-effect of therapy was considered a no-go for most potential cannabinoid-based therapies. This is part of the reason why much research has targeted enzymes such as FAAH to put endocannabinoids to work: these approaches are not intoxicating. An attractive early prospect therefore was to focus not on activating CB$_1$ receptors but on CB$_1$ antagonists that would avert the issue of intoxication altogether. A commercially attractive target presented itself since it was established that cannabinoids enhance appetite [30]. Companies such as Sanofi Recherche recognized the potential for profit from a drug that produced the opposite of "the munchies": weight loss. Sanofi developed its flagship compound SR141716, dubbed rimonabant (Figure 1) and marketed as Accomplia, as well as several follow-on compounds.

When researchers at Sanofi pondered what they might do with their flagship CB$_1$ antagonist rimonabant, they encountered a clear commercial choice. When activated, CB$_1$ receptors stimulate appetite and motivate eating behavior by acting on orexin melanin, concentrating hormone signaling in the hypothalamus [31,32]. They also enhance the sensitivity to sweet taste [33,34]. Opposing this system promised a novel tool to reduce appetite and promote weight loss. And indeed, early studies were encouraging; in clinical trials, patients reliably lost weight. The amount lost varied from individual to individual, but averaged 8–10 lbs [35]. Attractive from a commercial standpoint, the effects required continued treatment; if patients stopped taking the compound, the weight returned.

Sanofi, based in France, considered the potential market to be enormous, particularly in the US where obesity was, and continues to be, a major health concern. With 160 million Americans overweight or obese [36], the potential market for an effective weight-control drug is, by any standard, considerable. Rimonabant was approved in Europe in 2006 as a treatment for weight loss and was actively considered for approval by the FDA (but rejected based on concerns about aversive psychoactive side-effects).

### 3.2. The Clinical Trials

After a series of preclinical studies, four clinical trials assessed rimonabant's efficacy in reducing weight, as well as several cardiovascular and metabolic conditions associated

with obesity. Rimonabant in obesity (RIO)-Europe and RIO-North America, tested for weight loss in overweight and obese participants over a 2-year period [37–39]. RIO-Lipids and RIO-Diabetes expanded these goals to include cardiovascular risk in high-risk patients with dyslipidemia, metabolic syndrome, and type 2 diabetes [35,40]. The one-year follow-up saw significant reductions in weight with rimonabant. But those taking rimonabant also saw positive changes in their levels of plasma C-reactive protein, HDL cholesterol, triglyceride, adiponectin, HbA1c, fasting glucose and insulin, as well as a measure of insulin resistance (Homeostatic Model Assessment for Insulin Resistance (HOMA-IR)) [35,37–40]. The 20 mg/kg dose also reduced LDL cholesterol and the prevalence of those meeting the criteria for hypertension and metabolic syndrome at the one-year mark [35,40]. The observations for weight loss and improvements in lipogenic and glycemic profiles remained consistent at the second follow-up one year later for the RIO-Europe and RIO-North America studies [38,39]. Rimonabant was therefore poised to enter the stage as a first-in-class effective treatment to reduce weight and, on the strength of these findings, it received approval in Europe. But, the clinical trials also revealed a dysphoric side-effect profile. This would be the undoing of rimonabant and, in 2008, it was withdrawn worldwide [41].

### 3.3. The Fall

Besides the truly striking metabolic efficacy of rimonabant, reports appeared of a darker side to the weight-loss treatment: was it possible that the drug was producing not only the opposite of 'the munchies', but also the opposite of euphoria? Clinical trials reported nausea and dizziness, but also increased depression, anxiety and the specter of suicides sounded the death-knell for rimonabant as a weight-loss therapy and several other indications. Approval in Europe was withdrawn in 2008 based on postmarketing surveillance, and the requests for approval were withdrawn in the US and elsewhere. Other companies with 'me-too' CB$_1$ antagonists such as Merck's taranabant [42] and Pfizer's CP945598 [43] quietly terminated or shelved their clinical trials. Clinical trials underway for rimonabant to help smokers quit, though promising, were also terminated. Pharmaceutical companies are generally conservative, reluctant to introduce drugs for an unproven target for fear of unexpected side-effects. The experience with first-in-class rimonabant had proven to be a worst-case scenario.

## 4. A Search for Alternatives

Nearly fifteen years have passed since the end of this chapter. The experience with rimonabant represented an enormous setback and even today there are no clinical trials underway in the US with the specific goal of using a conventional CB$_1$ antagonist for therapeutic ends. Most active cannabinoid-related clinical trials involve the use of enzyme blockers, agonists or, more recently, minor phytocannabinoids. But the story of CB$_1$ receptor antagonists as therapeutics did not end here. If anything, obesity has worsened. And so, researchers have pursued alternative pharmacological strategies, ones that might avoid the aversive side effects. Moreover, the clinical trials for rimonabant and related compounds have revealed other cardiac and metabolic benefits [39]. And a few additional potential uses for a CB$_1$ antagonist have surfaced. We will briefly review potential therapeutic applications for CB$_1$ antagonists, then some novel pharmacological strategies that are being pursued.

### 4.1. A Peripheral Interest

Rimonabant was clearly effective for moderate weight loss in human subjects and had attractive metabolic effects. Was there any hope for a cannabinoid antagonist or was the entire class of compounds doomed? Might a CB$_1$R antagonist be developed that somehow avoided the dysphoria? Several potential strategies were considered but the most compelling arose from the observation that not all of the effects of rimonabant were due to actions in the CNS. The metabolic effects along with CB$_1$R expression in peripheral tissues such as adipocytes, the liver, and components of the GI tract already pointed to CB$_1$R

roles in the periphery [44,45]. Moreover, when $CB_1Rs$ were antagonized in the periphery but not the CNS, energy consumption was 'normalized' and the ensuing weight loss was independent of food intake [46]. The changes in the metabolic profiles of RIO-study subjects were twice those expected from the weight loss alone, suggesting that rimonabant had a peripheral effect on glucose, lipid and insulin metabolism [35,39,40].

These observations suggested that a $CB_1R$ antagonist that either did not penetrate into the CNS or that did not act as an inverse agonist might therefore still prove useful. In the 15 years since rimonabant's fall from grace, researchers have pressed on to understand how cannabinoids interact with various peripheral players in the metabolic process. The subject is complex as $CB_1Rs$ turned out to play a multifaceted role in the development of many pathologies associated with metabolic syndrome (see, e.g., [47] for review). While our understanding is incomplete, several areas are of particular interest, particularly relating to lipogenesis. Adipocytes express $CB_1R$, especially in mature adipose tissue, and their levels increase with obesity [48,49]. Blocking $CB_1$ receptors reduces fat synthesis and storage, lipoprotein lipase activity, and hepatic fatty acid synthesis [44,46,50], suggesting that the endocannabinoid system regulates the cellular machinery of fat cells [49,51]. Furthermore, the blockade of adipocyte $CB_1Rs$ by rimonabant was reported to increase the secretion of adiponectin, a hormone that is significantly curtailed in models of human and murine obesity [35,40,52], and which promotes free fatty acid oxidation, body weight reduction, improves hyperglycemia and hyperinsulinemia, and reverses insulin resistance in obese animals [53]. The importance of adipocyte $CB_1Rs$ is also corroborated by the finding that selective ablation of $CB_1Rs$ in fat cells is sufficient to normalize body weight in obese mice [54].

$CB_1R$ also plays a role in the liver. Hepatocytes, key players in the metabolic process, appear to produce 2-AG [55] and $CB_1R$ activation in the liver, elicited by, for example, a high-fat diet, stimulates the expression of genes involved in fatty acid synthesis such as transcription factor SREBP-1C, acetyl CoA carboxylase-1, and fatty acid synthase [56]. This lipogenic response can be blunted by $CB_1R$ antagonists, with positive implications for treating not only obesity, but also fatty-liver disease. Moreover, $CB_1Rs$ in the liver bring about hepatic insulin resistance in a diet-dependent manner [57] and promote leptin resistance [58], thus providing yet another therapeutic rationale for blocking hepatic $CB_1Rs$.

Such reports of peripheral metabolic effects notwithstanding, the greatest challenge for a new cannabinoid-based therapeutic for weight loss may simply be that the landscape has changed. At the time of approval, rimonabant faced little serious competition for weight loss, but this would not hold true today. The GLP1 receptor agonist semaglutide was recently approved as a therapy for weight loss and several other classes of drugs are in clinical trials (reviewed in [59,60]). If the results reported thus far hold true, overweight patients can hope to lose ~12% of their weight, or more than twice what was reported for rimonabant. If GLP1 agonists prove safe and effective, then they will be the bar against which a successor to rimonabant will be judged, though it may prove attractive as part of a combination therapy, since peripheral $CB_1R$ agonists appear to provide beneficial metabolic effects that are independent of weight loss or glycemic control [47]. If one were to identify a mechanistic basis for why some patients experienced prodigious weight loss over others, this might also serve as an attractive direction of research.

The potential use of a $CB_1R$ antagonist in the cardiovascular system was also explored as a therapeutic target, with mixed results that are surveyed in several excellent reviews (e.g., [61]).

### 4.2. A Therapy for Substance Abuse

The first report of $CB_1R$ knockout mice noted that $CB_1R$ deletion impacted opiate tolerance [62]. Might the cannabinoid signaling system, with its own abuse liability, be helpful for opiate abuse and might this apply to other drugs of abuse (reviewed in [63])? $CB_1R$ deletion or blocking by rimonabant was found to be helpful not only for quitting opiates, but for a spectrum of drugs of abuse, including cocaine [64], alcohol [65], and nicotine [66]. We will

focus on smoking cessation since the development of rimonabant as a therapeutic for smoking addition was actively pursued in clinical trials by Sanofi. The need was—and remains—clear: smoking tobacco is still one of the most prevalent, avoidable causes of death [67]. It is also highly addictive: 80% of smokers who attempt to quit relapse within the first month of abstinence [68]. To understand how $CB_1$ receptors might help smokers quit, we need to understand the underpinnings of dependence in smoking. Nicotine is a potent agonist at the eponymous nicotinic acetylcholine receptors (nAChR) [69]. In the brain, these receptors excite neurons and elicit the release of multiple neurotransmitters; importantly here, this includes the release of dopamine which mediates the mild sense of pleasure experienced by smokers by activating the mesocorticolimbic reward pathway [70] (Figure 4). This reward pathway is a critical neuronal underpinning of drug addiction, and its relationship with nicotine has been studied intensely for decades (reviewed in [71]). Nicotine-stimulated dopamine release in the ventral tegmental area leads to dopamine release in the nucleus accumbens (NAc) and nucleus of the stria terminalis [72,73]. How might $CB_1$ receptors help? They are found presynaptically at key synapses in the mesocorticolimbic system, including both GABAergic and glutamatergic afferent neurons that regulate dopamine release (Figure 4B, [74,75]). $CB_1$ receptors on GABAergic inputs inhibit the release of GABA, relieving their inhibition and enhancing dopamine release (Figure 4). Rimonabant would be expected to oppose this, reducing the ability of nicotine to produce a pleasurable effect [76]. A study of nicotine-induced dopamine release in the NAc found this effect for rimonabant vs. nicotine [66] and ethanol [66] but a separate study did not see a comparable result for heroin [77] even though it found that rimonabant reduced self-administration.

Reward is central to the initiation of drug use, but drug dependence is about more than reward. Withdrawal symptoms in the wake of abstinence can be a powerful inducement to relapse. Rimonabant did not induce withdrawal in nicotine-dependent mice, though the same study found that $CB_1R$ activation ameliorated withdrawal symptoms [78]. Learned environmental cues can also contribute to dependence and are linked to dopamine release [79]. This has been demonstrated for nicotine and is countered by rimonabant, which reduced cue-associated relapse in nicotine-dependent rats [80–82]. There is evidence that cue-associated relapse occurs through the modulation of the impact of reward-related memories [83]. $CB_1$ receptors mediate long-term plasticity in several brain regions central to the formation and evaluation of memories (i.e., hippocampus, amygdala, prefrontal cortex [23]) and $CB_1R$ inhibition improves some aspects of memory [84] but the mechanism by which rimonabant reduced cue-associated relapse remains uncertain.

Lastly, cannabinoids also impact the motivation to seek out opioids and psychostimulants through a mechanism that is independent of dopamine release in the nucleus accumbens. This has less relevance for nicotine, but may apply for other important drugs of abuse, and potentially for food craving. The basis for this is still a matter of speculation but may involve $CB_1$ receptors in the prefrontal cortex that integrate and bind sensory, emotional and hedonic inputs (discussed in [63]).

These findings led to consideration of rimonabant as a potential therapy to help smokers quit. Experiments in animal models of addiction and withdrawal proved promising: rimonabant reduced dopamine release in the nucleus accumbens and animals were less likely to self-administer nicotine even when presented with associated cues (e.g., [66]). Thus encouraged, researchers initiated a series of five clinical trials collectively named STRATUS (Studies with Rimonabant and Tobacco Use) for treating nicotine dependence in those motivated to quit. Pooled analysis of these studies found that those who took rimonabant (20 mg) had a 50% higher chance of maintaining abstinence [85]. A CIRRUS study, not affiliated with STRATUS, saw even better outcomes (39% vs. 21%, a ~2-fold improvement) when rimonabant was used in combination with nicotine replacement therapy [86]. Most participants (>60%) were resistant to the treatment, which included weekly counseling, but it had the side-benefit of averting the weight gain that is frequently associated with quitting smoking. At the time, smokers had few options aside from nicotine replacement therapies

and anti-depressants [87] but varenicline, a nicotinic receptor agonist developed for this purpose, has since been reported to yield superior outcomes (2.9-fold improvement) [88].

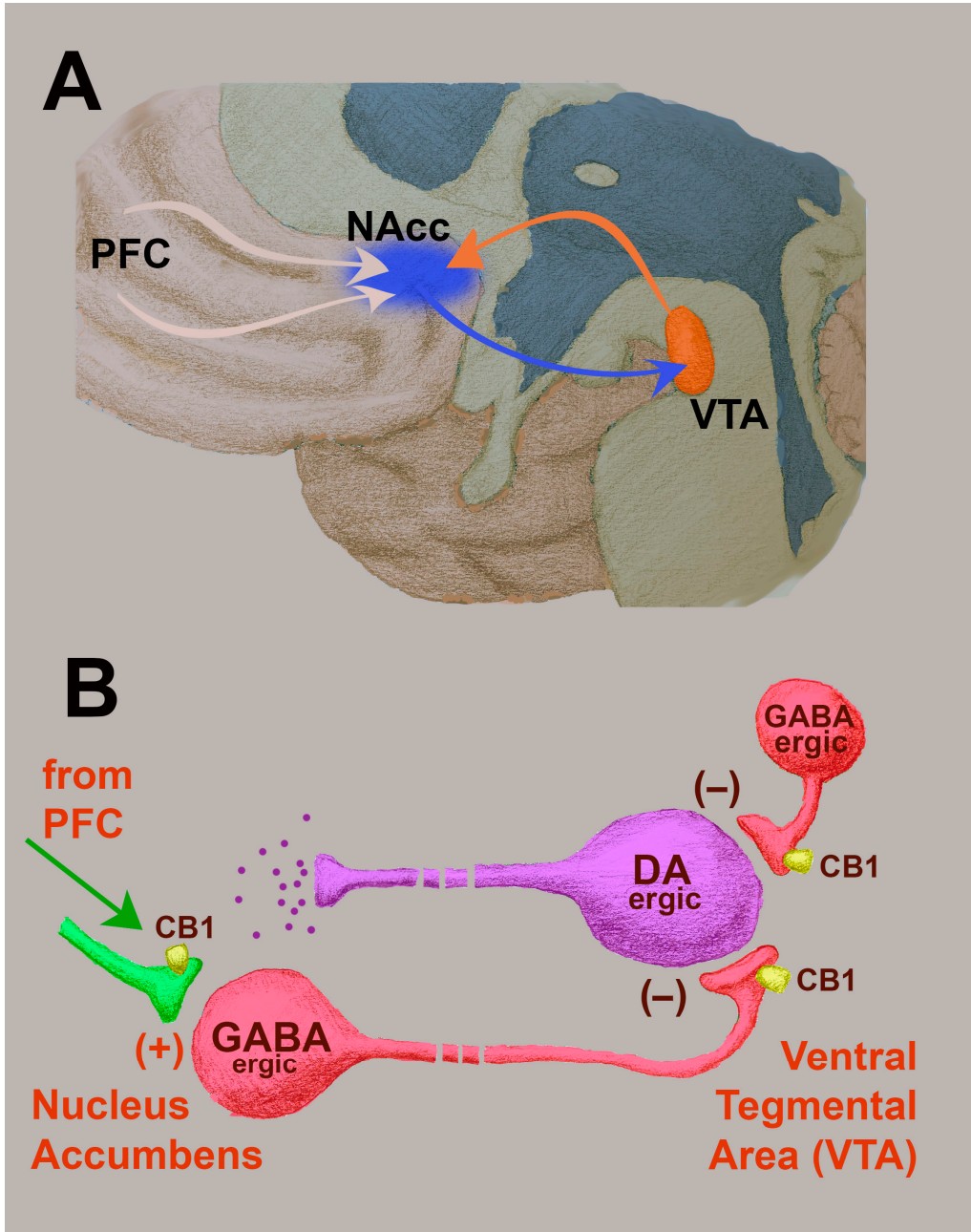

**Figure 4.** $CB_1R$ in reward circuitry. (**A**) The nucleus accumbens (NAc) receives excitatory input from the prefrontal cortex (PFC). The NAc also sends inhibitory projections to the dopaminergic ventral tegmental area (VTA), reducing release of dopamine (DA). (**B**) In the NAc, cannabinoid receptor type 1 ($CB_1$) on PFC inputs reduce the stimulation of GABAergic NAc neurons. By reducing GABA release onto dopaminergic (DAergic) VTA neurons, $CB_1Rs$ increase DA release. $CB_1$ receptors are also present in the VTA on GABAergic inputs. Here too, $CB_1R$ activation relieves inhibition of DAergic neurons, increasing DA release.

A post hoc evaluation of three unpublished trials found that the side effect profile was not as pronounced as those reported for studies of the drug for the purpose of weight loss but did include anxiety, nausea, diarrhea and vomiting [85]. One favorable consideration is likely duration of treatment. Rimonabant treatment for weight loss would ultimately

be life-long (since weight returned after patients stopped using rimonabant), while treatment for smokers might not be needed after patients had passed through a window of vulnerability. In 2006, the FDA gave Sanofi a non-approvable letter for its use in smokers (but an approvable letter for weight loss). The clock ran out on rimonabant before it could address FDA concerns. After the European Medicines Agency withdrew approval, Sanofi retracted its NDA [89] and did not pursue the research further. Research on the use of $CB_1$ antagonists for dependency on nicotine and other drugs of abuse drastically declined over the following decade. Interest in the approach has not evaporated and some studies are exploring pharmacological alternatives to rimonabant [90] that may avoid the psychiatric side-effects, as will be discussed in the last section. For instance, the peripheral $CB_1R$ antagonist JD5037 was found to reduce ethanol drinking in wild type, but not in $CB_1R$, ghrelin or ghrelin receptor knock-out mice [91], thus revealing a hitherto unrecognized gut-brain axis in alcohol abuse.

### 4.3. Cannabis Toxicity

When emergency providers encounter a victim of opiate overdose, they turn to naloxone, a competitive antagonist at the mu opioid receptor that has served as an antidote to opiate toxicity for 50 years. There is no comparable treatment for cannabis overdose even though the need is there. Emergency rooms are seeing a spike in cannabis-related visits. This is partly due to the spread of cheap and extremely potent synthetic cannabinoids that can cause serious neurological and cardiovascular complications, and roughly a dozen deaths per year in the US. But the bulk of these visits—nearly half a million per year in the US—are due to overdose of cannabis, especially in regions that have legalized recreational cannabis [92]. Absent an antidote, treatment options are mostly limited to sedatives, with their own risks, to 'wait out' the overdose. For acute single-use reversal of toxicity in an emergency setting, the benefits of using a compound such as rimonabant may outweigh the risks. The real question is whether rimonabant has the properties of a good antidote: simple administration suitable to an emergency setting and rapid action. The preferred route would be intramuscular injection or nasal spray and the effect onset should ideally be under five minutes. The lack of any published data on this subject may be an indication that rimonabant does not meet these criteria or—less likely—that no one has tested rimonabant for this purpose. It will be interesting to see whether the biased $CB_1R$ inhibitor AEF0117, which proved to be efficient in treating cannabis-use disorder in Phase 2a trials [93], will emerge as potential treatment for cannabis intoxication.

At the time that rimonabant was in clinical trials, there were limited applications for a $CB_1$ antagonist beyond weight loss and drug addiction. This has changed somewhat as we have developed a more thorough understanding of the many roles played by cannabinoid receptors in the body. Cannabinoid effects on metabolism are a potentially rich vein that will spur new lines of clinically motivated research. Doubtless, other applications for cannabinoid antagonists will become apparent in time. At that time, new pharmacological tools will be available, as discussed in the next section.

## 5. Novel Pharmacological Strategies

### 5.1. Peripherally Restricted Inverse Agonists

As mentioned earlier, there is evidence that at least some weight-loss and metabolic benefits are to be had by blocking $CB_1$ receptors outside of the CNS. Researchers seized on this to focus on the development of peripherally restricted $CB_1$ receptor antagonists, i.e., those that do not cross the blood–brain barrier. Often chemists face the opposite challenge, of modifying compounds to facilitate their entry into the brain, the portals of which are guarded jealously by the various channels and pumps that make up the blood–brain barrier. Endocannabinoids and phytocannabinoids are as a rule lipophilic, and readily cross through this barrier. Determined, chemists soon developed compounds, generally variants of rimonabant and other known antagonists that they were modified to be less lipid-soluble. Results with peripherally restricted $CB_1R$ antagonists have proved promising

as they do not induce withdrawal (e.g., [94]), but compounds such as JD5037 reduce food intake and body weight in mice with diet-induced obesity through the normalization of hyperleptinemia and restoration of central leptin sensitivity [45,95,96]. Moreover, a peripheral CB$_1$ blockade has the potential to significantly delay the progression of β-cell loss [97] and diabetic nephropathy [98] independently of food intake and body weight reduction.

### 5.2. Neutral Antagonists

Another proposed strategy was the development of neutral, or 'silent', antagonists. The premise was that rimonabant was not merely an antagonist but an inverse agonist. There is evidence that in some settings, CB$_1$R is inactive under baseline conditions but in others there is a 'tonic' activity that is independent of ligand binding. For instance, CB$_1$R inverse agonists alone will increase gastrointestinal motility [99], thus implying tonic CB$_1$R activity on vagal terminals. On the other hand, CB$_1$R inverse agonists typically do not increase core body temperature [100] while hypothermia is one of the classical effects of pharmacological CB$_1$R activation. This suggests that CB$_1$Rs that mediate the hypothermic effect of cannabinoids are not tonically active, and that different sites of the body have a different 'endocannabinoid tone'. Moreover, some data suggest that this 'tone' may vary among individuals and under pathological conditions such as insulin resistance and obesity [101]. In principle, such a difference might provide a mechanistic basis to explain why some patients experienced considerable weight loss in response to rimonabant while others did not.

Taking the example of anxiety, if the cannabinoid system is partially active to reduce anxiety, then reversing this to zero with an inverse agonist would result in depression. Based on this reasoning, a neutral antagonist would maintain signaling at the partial tonic state. Several planets have to be in alignment for this to work. The circuitry controlling appetite needs to *not* have ligand-independent tonic activity while the circuitry impacting moods that were problematic for rimonabant do. Several neutral antagonists have been described and tested in the context of weight loss and/or smoking (e.g., VCHSR [102]; AM4113 [103]). Promisingly, AM4113 reduced food intake and food-reinforced behavior without causing nausea or increased responses to fear conditioning or anxiety [104–106]. Finally, AM6545, a CB$_1$R antagonist that is both neutral and non-penetrant, improved plasma and liver lipid parameters, adiposity and body weight in diet-induced obese animals while lacking detectable behavioral side effects [95].

### 5.3. Biased Antagonists

In the pursuit of safer CB$_1$R blockers, Cinar and co-workers developed a β-arrestin2 (arrestin3) biased orthosteric antagonist, named MRI-1891, that does not inhibit CB$_1$R-mediated G$_i$ signaling [107]. This compound, unlike rimonabant, interacts with nonpolar residues close to the N-terminus of the receptor that is likely the molecular underpinning of biased antagonism. Prominently, MRI-1891 improved muscle insulin resistance and reduced body weight in diet-induced obese mice while displaying no anxiogenic activity even at very high doses with partial brain CB$_1$R occupancy. Also, this compound proved to be effective in ameliorating diabetic nephropathy [108]. Another biased compound, the pregnenolone derivative AEF0117, which selectively blocks CB$_1$R-mediated MAP kinase signaling without affecting cAMP levels and is efficacious in the treatment of cannabis use disorder, has no detectable adverse neuropsychiatric effects, despite high brain penetrance [93]. All in all, biased antagonism may be yet another strategy to overcome undesired neuropsychiatric side effects associated with a traditional, central CB$_1$ blockade while retaining some therapeutic effects of CB$_1$R antagonism.

### 5.4. Turning off the Tap—Blocking Endocannabinoid Synthesis

In principle, another strategy would be to develop blockers for the synthetic enzymes for either of the endocannabinoids. In the case of 2-AG, this would be either of two dia-

cylglycerol lipases (DAGLa and DAGLb) [11]. DAGLa appears to have a more prominent CNS role, while the two share roles in the rest of the body [109,110]. For anandamide, this was long an open question, but it is likely that NAPE-phospholipase D (NAPE-PLD) is responsible for synthesizing anandamide [12]. Pharmacological tools for these enzymes have been limited, but some DAGLa/b-selective compounds have been reported (e.g., KT109 [111]) and, lately, a blocker for NAPE-PLD [112]. Targeting such an enzyme may offer greater specificity and selectivity than a blanket blockade of all CB1 receptors, especially if the enzyme of interest has a more limited distribution. But, enzyme blockers may come with unexpected consequences. For example, altering 2-AG metabolism has been found to also have profound effects on the arachidonic acid cycle and on prostaglandin synthesis [109,113]. This approach is still in its infancy.

*5.5. Negative Allosteric Modulation*

As previously noted, one of the main challenges in developing cannabinoid therapeutics is the near ubiquity of $CB_1$ receptors. As a result, there is a strong interest in developing alternatives that offer more selective targeting. One strategy has been to develop allosteric modulators of $CB_1Rs$. The idea is that most receptors have not only their classical 'orthosteric' site, but also at least one secondary 'allosteric' site. In principle, a ligand that binds the allosteric site would modulate the signaling of this receptor by the endogenous ligand. A negative allosteric modulator (NAM) would inhibit the endogenous signaling, while a positive allosteric modulator (PAM) would enhance that signaling. Allosteric modulators are not a new concept—benzodiazepines and barbiturates are PAMs at GABA-A receptors—but allosteric modulators for cannabinoids were not described until 2005 [114]. Research has continued into allosterics, with particular interest in $CB_1$ PAMs for alleviation of pain [115]. A CB1 NAM promises the possibility of dialing down existing signaling only at receptors that are being activated endogenously. This may be subject to the same pitfalls as traditional competitive antagonists. But, the NAMs described thus far (e.g., [116]), have shown considerable 'biased antagonism'. GPCRs activate multiple intracellular signaling pathways; biased signaling means that a given ligand differentially affects these pathways. In some cases, they even have a mix of activating and inhibiting effects. This may prove advantageous if it can be determined, for instance, that desirable effects occur via a particular pathway, similarly to that reported for MRI-1891 (see above). This advantage may also apply to conventional antagonists.

*5.6. Phytocannabinoids*

The subject of NAMs brings us to the last group of compounds, the phytocannabinoids that started this journey. In some sense, the cannabinoid field has come full circle. After the initial flourish of phytocannabinoid research in the 1970s, efforts focused first on the newly identified receptors and their endogenous ligands, and then on defining the enzymatic players in their synthesis and metabolism. Lately, however, there has been a rebounding interest in plant cannabinoids. This has been due in part to the changing legal status in some countries where companies have zealously embraced all things cannabis, but also because of the striking effects of one cannabinoid that was long ignored. Though THC and CBD are generally present in comparable quantities in the plant, CBD long remained in the shadows, often referred to as the inactive or at least a non-psychoactive plant cannabinoid. This picture was based in large part on early studies that showed that CBD did not activate $CB_1$ cannabinoid receptors [117], studies that missed allosteric binding to $CB_1R$. CBD is likely a negative allosteric modulator at $CB_1$ receptors [118] and it has been demonstrated that CBD blocks the effects of equivalent concentrations of THC, for instance, in the regulation of ocular pressure [119] and salivation [120]. But, the salutary effects for the control of seizures are likely to occur through another receptor such as GPR55 [121]. CBD has been investigated in animals for potential effects on weight, with mixed results (reviewed in [122]). The embrace of CBD by the public and its FDA approval as a treatment for a form of epilepsy has led to a re-appraisal of the 100+ other phytocannabinoids. Several

dozen early-stage clinical trials are underway, none of these for CB$_1$ antagonist properties, but it is noteworthy that the cannabinoid-focused GW Pharmaceuticals has a patent filing that lists tetrahydrocannabivarin (THCV) as a neutral antagonist [123].

## 6. Conclusions

The endocannabinoid system is found throughout the CNS and the body where it impacts many important physiological processes (summarized in Figure 5). Expectations were high that targeting cannabinoid receptors would prove therapeutically beneficial; pharmaceutical companies labored long to develop a therapy. Alas, the experience with first-in-class cannabinoid type-1 receptor (CB$_1$R) antagonist rimonabant as a therapy for weight loss is a now-classic cautionary tale of the perils of drug development. The outcome dealt a blow to those pursuing therapies involving CB$_1$R antagonists. Even the most commercially compelling application of rimonabant—weight loss—has now been partially eclipsed by drugs with different mechanisms of action and greater effect. Still, blocking CB$_1$ receptors results in intriguing metabolic effects, some of which appear to occur outside the CNS. Moreover, recent years have seen a startling change in the legal status of cannabis, accompanied by a popular embrace of 'all things cannabis'. These changes combined with new pharmacological strategies and diligent medicinal chemistry may yet see the field to some measure of fulfillment of its early promise.

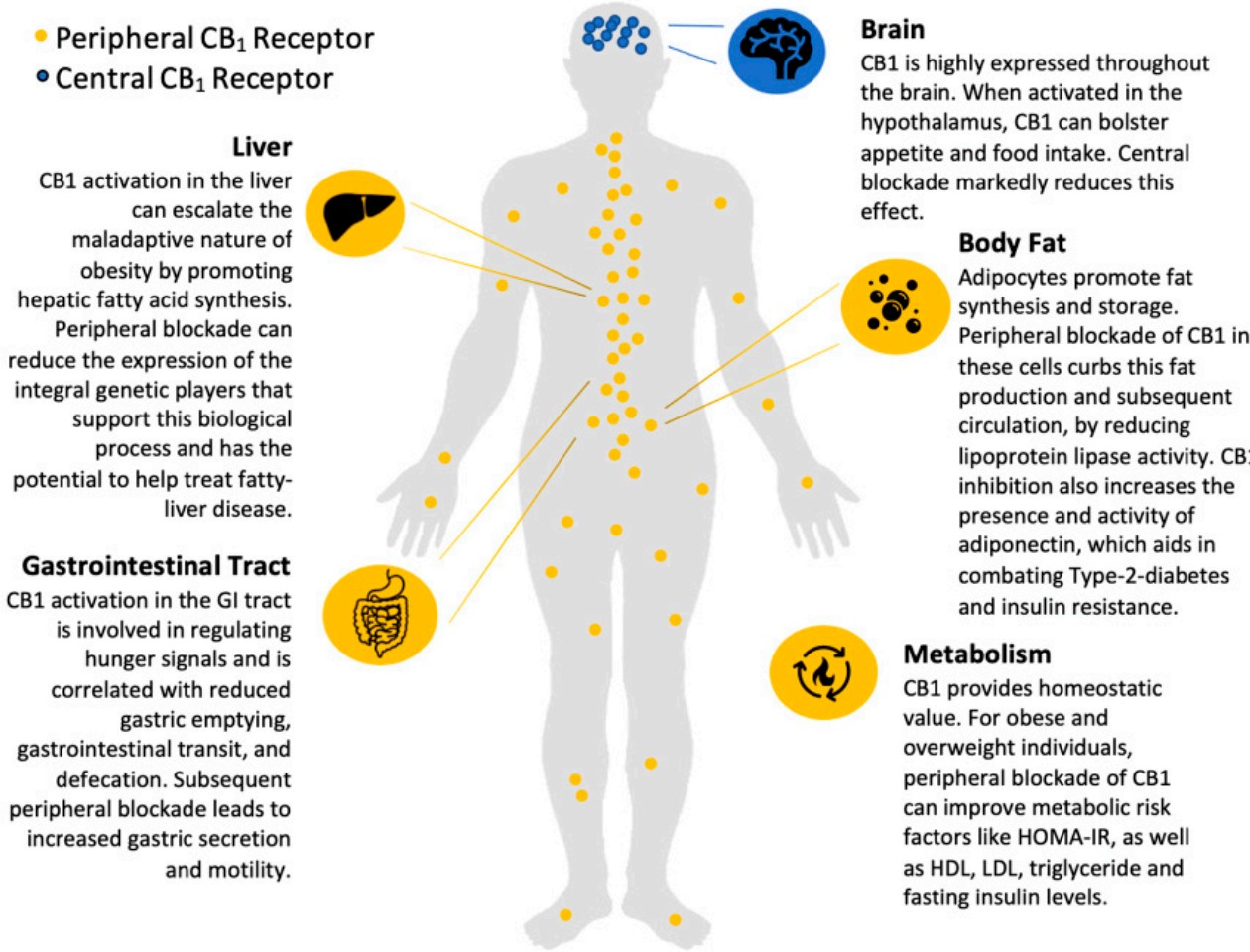

**Figure 5.** Central and peripheral metabolism-related targets for a CB$_1$ antagonist. Cannabinoid type 1 (CB$_1$) receptor expression is high throughout the brain and is also distributed throughout the periphery. Abbreviations: gastrointestinal tract (GI), Homeostatic Model Assessment for Insulin resistance (HOMA-IR), high-density lipoprotein (HDL), and low-density lipoprotein (LDL).

**Author Contributions:** Conceptualization: A.S. Writing—Original Draft Preparation: A.S., T.B.-B. and G.S. Writing—Review and Editing A.S., G.S. and T.B.-B. All authors have read and agreed to the published version of the manuscript.

**Funding:** This research received no external funding.

**Institutional Review Board Statement:** Not applicable since this review article did not include the use of experimental subjects.

**Informed Consent Statement:** Not applicable.

**Data Availability Statement:** Not applicable since manuscript does not contain experimental data.

**Acknowledgments:** We thank Huei-Ying Chen for the use of her micrograph (Figure 2).

**Conflicts of Interest:** The authors declare no conflict of interest.

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
