# Peer review of "Requiem for Rimonabant: Therapeutic Potential for Cannabinoid CB1 Receptor Antagonists after the Fall"

_ddc, doi:10.3390/ddc2030035_

Round 1
Reviewer 1 Report
This is a well-written and authoritative review of the fall from grace of the CB1 receptor antagonist (/ inverse agonist) rimonabant as an anti-obesity agent. The review contains an appropriate background and detail of the high-profile withdrawal of rimonabant during post market surveillance and some good insights into the reasons behind this failure.
There is also appropriate discussion of the place for CB1 receptor antagonists in the general field of weight loss therapy and the area of potential alternative uses for such agents; for example, in shorter term use for smoking cessation and as antidote treatments. There are also solid ideas around combatting diseases associated with a restriction of drug to peripheral compartments.
Overall, I have just a few suggestions for improvement.
1. It seems useful to explain the meaning of “after the Fall”. This phrase is fine, but some readers may apply different meanings, better to explain early on in the article.
2. The background information on the endocannabinoid system (ECS) and signalling is appropriate, however, this section may benefit from a clear descriptive on the retrograde (vs anterograde) nature of cannabinergic signalling. In this regard, do the authors believe that any of the adverse effects of rimonabant can be ascribed to this somewhat unique signalling mechanism?
3. The ideas around the pharmacological mechanism of action of rimonabant are useful, but could be developed further. For example, the inverse agonist nature of rimonabant is likely a key factor. The idea to develop neutral (and/or biased) antagonists that may avoid potential adverse effects on ‘endocannabinergic tone’ in the CNS is good. In this regard, an expansion of the discussion around EC tone may be useful. For example, is it known to what extent EC tone differs between subjects? Is it possible that those adversely affected by rimonabant have an inherently ‘low tone’ that is suppressed to what may be dangerous levels, and could lead to the unwanted side effects detailed?
4. The proposal that negative allosteric modulators may be useful in weight loss is of interest; as the authors point out, this is one of the (several) proposed mechanisms of action of cannabidiol. Is there any evidence that CB1 NAMs and/or CBD support weight loss?
Minor points
5. Ln 128 – explain the meaning of “other subjects”.
6. Ln 420 Some readers may not understand the term “spigot”- perhaps “tap” is better?
7. Several references are not formatted correctly.
Author Response
We thank the reviewers for their close reading of the manuscript and for the constructive nature of the comments. Moreover, some of the comments were thought provoking and certainly contributed to the overall quality of the review. Please see our point-by-point replies to the reviewer comments.
- It seems useful to explain the meaning of “after the Fall”. This phrase is fine, but some readers may apply different meanings, better to explain early on in the article.
A: We added mention of a ‘fall from grace’ to the abstract that should help provide context for the term, albeit without an explicitly biblical connotation.
- The background information on the endocannabinoid system (ECS) and signalling is appropriate, however, this section may benefit from a clear descriptive on the retrograde (vs anterograde) nature of cannabinergic signalling. In this regard, do the authors believe that any of the adverse effects of rimonabant can be ascribed to this somewhat unique signalling mechanism?
A: We have added some additional explanation/clarification (lines 101-103).
It seems likely that alteration of some form of cannabinoid-mediated retrograde signaling underlies the dysphoric effects of rimonabant – presynaptic CB1 receptors are present in several parts of the reward circuitry of the nucleus accumbens (as depicted schematically in Fig. 4). We also added a sentence to draw attention to the somatodendritic CB1R population, which may convey biological effects different from those of the presynaptic receptors (~lines 106-108).
- The ideas around the pharmacological mechanism of action of rimonabant are useful, but could be developed further. For example, the inverse agonist nature of rimonabant is likely a key factor. The idea to develop neutral (and/or biased) antagonists that may avoid potential adverse effects on ‘endocannabinergic tone’ in the CNS is good. In this regard, an expansion of the discussion around EC tone may be useful. For example, is it known to what extent EC tone differs between subjects? Is it possible that those adversely affected by rimonabant have an inherently ‘low tone’ that is suppressed to what may be dangerous levels, and could lead to the unwanted side effects detailed?
A: We have added some additional discussion of eCB tone (line ~385-393).
- The proposal that negative allosteric modulators may be useful in weight loss is of interest; as the authors point out, this is one of the (several) proposed mechanisms of action of cannabidiol. Is there any evidence that CB1 NAMs and/or CBD support weight loss?
A: This is an interesting point though CBD, with its many likely targets, may not be an optimal test case. We have updated the text to include more information about this (line ~470).
Minor points
- Ln 128 – explain the meaning of “other subjects”.
- Ln 420 Some readers may not understand the term “spigot”- perhaps “tap” is better?
- Several references are not formatted correctly.
A: We have updated the text to address these suggestions/concerns.
Reviewer 2 Report
The article by Bosquez-Berger et al. it is very important and impactful because it clarifies and highlights the characteristics (pros and cons) of a molecule that has made the history of the last few decades in the field of cannabinoids and the potential medical applications related to them. In my opinion it can be published after the minor revisions indicated below.
Figure 1. Add the structure of AEA.
Novel pharmacological strategies. The authors forgot to mention a historical article, which opened the way to possibilities not known until now. Therefore, the article LoVerme et al Bioorg Med Chem Lett 2009, 19, 639-643 must necessarily be mentioned.
Author Response
We thank the reviewers for their close reading of the manuscript and for the constructive nature of the comments. Moreover, some of the comments were thought provoking and certainly contributed to the overall quality of the review. Please see our point-by-point replies to the reviewer comments.
Figure 1. Add the structure of AEA.
A: We have added this structure to figure 1.
Novel pharmacological strategies. The authors forgot to mention a historical article, which opened the way to possibilities not known until now. Therefore, the article LoVerme et al Bioorg Med Chem Lett 2009, 19, 639-643 must necessarily be mentioned.
A: We have added this reference (line 370).
Reviewer 3 Report
In a sea of reviews about the endocannabinoid system and related therapeutics, the review by Bosquez et al, distinguishes itself from the rest, which in itself is no mean feat! Apart from the eloquently-written, narrative account of the history of the cannabinoid therapeutics, the authors have also covered several important details which other reviews on this topic have routinely missed (such as the issue with FAAH blocker BIA-10-2474 to name a few). I also very much liked that the authors compared the efficacy of rimonabant to the current weight loss (potential gold standard) drug, semaglutide
The strongest point of the review also leads to some frailties. The narrative aspect of the review is great when it comes to giving a historical account of rimonabant. But when it comes to describing certain technical aspects of signaling or therapeutics, the narrative theme needs to take a back seat and the facts need to be supported slightly better by references. Below are a few comments for the authors-
1. Although the review's central theme is on CB1R antagonists, you mention about CBD in various parts of the review. While you do describe it as being a negative allostric modulator of CB1R (line 468), it is well-known to act on the TRPV channels (Lannottie et al., 2014). The latter infact has been been directly associated with its ability to lower seizure threshold (Gray and Whalley., 2020). So this needs to be mentioned if you want to include CBD in your review.
2. You mention about the biased antagonist in line 410. This is a 'peripheral' biased antagonist, so is the lack of anxiolytic effects due to it being peripherally restricted, or due to the selective Barr2 antagonism of the CB1R? Also are the deleterious effects of CB1R inverse agonist, rimonabant, to the reduction in Gi G protein recruitment, or both reduction of Gi and Barr recruitment?
3. The paragraph on neutral antagonists- Do you have references to support the claim that CB1R is exhibiting tonic activity in some conditions (such as anxiety) and not in others (such as the orexigenic response)?
4. Since the orexigenic circuits in the ARC are enriched in fenestrated capillaries, and devoid of a proper blood brain barrier, would it not be logical to just focus on peripherally selective CB1R antagonists, since they may still act on the ARC neurons (AGRP and POMC)? So as long as the antagonist is peripherally selective, would it make a considerable difference on the neuropsychiatric outcome if its neutral/biased antagonist or an inverse agonist? Side note- I did like the way the authors went about describing these pharmacological principles in a very intuitive kind of way.
5. I do not quite understand why there is a different paragraph for 'phytocannabinoids'? Would it not just fall under the NAM section for CBD, or neutral antagonist section for THCV?
6. The figure 5 you have is missing a significant portion of the text because its off center. The right side of it is completely cut off resulting in missing text.
7. While I do understand that the review is on CB1R antagonists, a major shift has been made to study the CB2R instead of the CB1R since you would be circumventing the issues with CB1R-associated neuropsychiatric effects. Do you suggest that this is a good alternative or are there any drawbacks to this line of research?
8. Minor comment-but you have to list the full form of all abbreviations in the legend for the figures, even if its commonly employed in the molecular signaling community (Ras Raf Mek Figure 3). Also, several abbreviations in figure 2 are not listed in the legend.
9. The section on potential use of rimonabant for substance abuse is a very interesting section! How does rimonabant compare in its efficacy for currently employed FDA drugs?
10. When it comes to the pathophysiological effects of CB1R activation, you do not necessarily go into much detail. You have a paragraph on the liver, and even less on the adipose tissue. Nothing much on the heart or vasculature. So I understand that it may be difficult to write about this at length, as it can disrupt the narrative theme, but you have to cite other recent reviews atleast that have covered this in greater detail. This would give the readers an opportunity to have a fundamental appreciation for CB1R-mediated effects before you jump into the therapeutics side of it.
11. Where do you see the largest market for the next generation of CB1R antagonists? Is it in obesity/weight loss still, or diabetes, fatty liver or metabolic syndrome or all of the above? Also what about cardiovascular function? Is there a scope for peripheral CB1R therapeutics in cardiovascular disease?
12. It is interesting that you do not describe the full signaling cascade of CB1R in the text, but do so in the figure only (figure 3). And this is not just in neurons, CB1R-mediated inhibition of cAMP generation is observed in virtually all cells where the receptor is expressed. Also several non-canonical signaling pathways exist for CB1R. Since you also talk about Barr2 in a later section, it becomes slightly important to actually go into detail and explain both the canonical (Gi coupled) and non-canonical signaling pathways (Barr1 and 2, and other G proteins) of CB1R.
Author Response
This appears to have been a copy/paste issue, perhaps because we were including a table as part of the reply to point No 9. We are including point 9 again here along with that table and our replies to the remaining concerns (10-12).
- The section on potential use of rimonabant for substance abuse is a very interesting section! How does rimonabant compare in its efficacy for currently employed FDA drugs?
A: Thank you, it does seem to have been a case of throwing the baby out with the bath water though as we noted in the text (line 317) at least one approved drug, varenicline, has a substantially greater likelihood of success in helping people to stop smoking. The efficacy of many FDA approved smoking cessation therapies has been extensively reviewed (Sigaly et al., 2002, Cahill et al., 2013) we have summarized those (see table below). All appear to be superior to rimonabant when comparing the odds ratios for smoking cessation but it is possible that further research might identify a more promising compound or that a compound might prove effective as part of a combination therapy. We took caution in reporting these values in the section, as the duration of treatment for each FDA approved therapy and rimonabant (Robinson et al., 2018) varied greatly and it is difficult to draw clear conclusions on comparative efficacy. We now also briefly mention the CB1 antagonist AEF0117, which shows substantial efficacy in treating cannabis use disorder in Phase 2a trials, as a potential future drug to facilitate the treatment of cannabis intoxication. (~lines 352 and ~410)
|
Smoking Cessation Therapy |
Odds ratio (OR) for cessation of smoking after treatment a |
|
Rimonabant *1 |
OR = 1.5 |
|
Varenicline 2 |
OR = 2.88 |
|
Bupropion 2 |
OR = 1.82 |
|
Nicotine gum 3 |
OR = 1.66 |
|
Nicotine patch 3 |
OR = 1.74 |
|
Nasal spray 3 |
OR = 2.27 |
|
Nicotine lozenges 3 |
OR = 2.08 |
|
*Pooled analyses from STRATUS-US and -EU a Duration of treatment and timing with which abstinence was measured post-treatment varied between each study. 1 Robinson et al., 2018 2 Cahill et al., 2013 3 Silagy et al., 2002 |
|
- When it comes to the pathophysiological effects of CB1R activation, you do not necessarily go into much detail. You have a paragraph on the liver, and even less on the adipose tissue. Nothing much on the heart or vasculature. So I understand that it may be difficult to write about this at length, as it can disrupt the narrative theme, but you have to cite other recent reviews at least that have covered this in greater detail. This would give the readers an opportunity to have a fundamental appreciation for CB1R-mediated effects before you jump into the therapeutics side of it.
We have added several references to flesh out this section. We also included recent reviews on the role of cannabinoids and CB1Rsin cardiovascular and metabolic diseases (please see answer to Q11 also).
- Where do you see the largest market for the next generation of CB1R antagonists? Is it in obesity/weight loss still, or diabetes, fatty liver or metabolic syndrome or all of the above? Also what about cardiovascular function? Is there a scope for peripheral CB1R therapeutics in cardiovascular disease?
With the availability of efficacious and safe incretin mimetics, the potential clinical niche for CB1R antagonist has certainly become narrower in the last decade. However, many of the beneficial effects of CB1R antagonists, such as improved insulin and leptin sensitivity, increased adiponectin secretion, protective capacity against pancreatic β-cell loss, fatty liver disease and glomerulopathy (see pertinent references in the MS), are independent of body weight reduction and blood glucose normalization. This suggests that peripheral CB1R agonists may be a good candidate for combination therapy with GLP-1 analogues and/or with antidiabetics, as CB1R blockade could provide the weight reduction and glycemic control independent protection from those pathologies. We touch on this possible therapeutic niche briefly in the MS (line ~256).
The role of cannabinoids in cardiovascular disease is an important and complex topic, one that is certainly beyond the scope of our review. Instead, we included excellent recent reviews. Regarding the activation of CB1Rs, it has adverse effects in the cardiovascular system, including tachycardia and decreased contractility of the myocardium. Moreover, cannabis users have somewhat poorer clinical outcome in myocardial infarction. On the other hand, rimonabant showed early promise in the treatment of cirrhotic hypotension (Batkai et al., Nat.Med., 2001), but research into its cardiovascular actions also suffered a blow once it was withdrawn from the market. (However, recent work of Hunyady and Szekeres on the interplay between the renin-angiotensin-aldosterone system and vascular endocannabinoids may be of interest to the Reviewer and cardiovascular researchers.). It was, nonetheless, well-documented in human trials that the favorable metabolic effects of rimonabant translate into substantially reduced cardiovascular risk (Van Gaal et al., Lancet, 2005).
- It is interesting that you do not describe the full signaling cascade of CB1R in the text, but do so in the figure only (figure 3). And this is not just in neurons, CB1R-mediated inhibition of cAMP generation is observed in virtually all cells where the receptor is expressed. Also several non-canonical signaling pathways exist for CB1R. Since you also talk about Barr2 in a later section, it becomes slightly important to actually go into detail and explain both the canonical (Gi coupled) and non-canonical signaling pathways (Barr1 and 2, and other G proteins) of CB1R.
We have added detail about the signaling pathways to the text (lines ~100-103) and a bit more information to the legend for figure 3.
Round 2
Reviewer 3 Report
Thank you for replying to all my comments. Tge replies are great, and the authors addressed all my concerns perfectly. I do have a slight concern. I listed 12 comments, but only 9 of them have been answered. Not quite sure if it's a technical glitch.
Author Response
Please find attached the full response to the reviewer's concerns including the last three points. We had included a table as part of response #9. This may have thrown off the paste function. We are also adding that reply as an attached document, just in case.
- Although the review's central theme is on CB1R antagonists, you mention about CBD in various parts of the review. While you do describe it as being a negative allostric modulator of CB1R (line 468), it is well-known to act on the TRPV channels (Lannottie et al., 2014). The latter infact has been been directly associated with its ability to lower seizure threshold (Gray and Whalley., 2020). So this needs to be mentioned if you want to include CBD in your review.
A: In our own work, we’ve looked into the effects of phytocannabinoids and their effects on TRP channels in DRGs (Straiker et al., 2021 https://pubmed.ncbi.nlm.nih.gov/34500785/). We did not look specifically at CBD in that paper but did have the occasion to evaluate the likely blood plasma concentration of CBD in a therapeutic setting in Straiker et al., 2018 https://pubmed.ncbi.nlm.nih.gov/29669714/). Reports such as the one by Iannotti (2014) make use of high (10-30uM) concentrations of CBD. The above paper has an interesting discussion of this but briefly we have pretty credible data for plasma concentrations of CBD from the Dravet’s studies and even with 20mg/kg daily doses the concentrations don’t exceed 1uM. For this reason we have left out mention of TRPV1 mechanism.
- You mention about the biased antagonist in line 410. This is a 'peripheral' biased antagonist, so is the lack of anxiolytic effects due to it being peripherally restricted, or due to the selective Barr2 antagonism of the CB1R? Also are the deleterious effects of CB1R inverse agonist, rimonabant, to the reduction in Gi G protein recruitment, or both reduction of Gi and Barr recruitment?
Indeed, MRI-1891 is both peripherally restricted and β-arrestin biased. However, with chronic high-dose MRI-1891 treatment, partial brain CB1R occupancy, comparable to that seen with acute 3 mg/kg rimonabant, could be attained (Fig.2b in Liu et al., 2021). Even then, no neuropsychiatric side effects were detected with this inverse agonist, strongly suggesting that Gi inhibition by rimonabant is necessary for anxiogenic actions. This notion is corroborated by the observation that very high doses of the neutral CB1R antagonist AM6545, producing measurable brain concentrations, lack detectable behavioral side effects (Tam et al., 2010) and that the brain penetrant CB1R biased inhibitor AEF0117, which inhibits CB1R mediated MAPK signaling but not the Gi pathway has no apparent central side effects (Haney et al., Nat.Med., 2023). These considerations are now discussed in more detail in the biased antagonist section.
- The paragraph on neutral antagonists- Do you have references to support the claim that CB1R is exhibiting tonic activity in some conditions (such as anxiety) and not in others (such as the orexigenic response)?
This is an intriguing question, and we are not aware of studies addressing it in a systematic manner. However, there is circumstantial evidence suggesting that some CB1Rs are in fact active tonically whereas other are activated `on demand`. We now discuss this matter in lines ~385-393).
- Since the orexigenic circuits in the ARC are enriched in fenestrated capillaries, and devoid of a proper blood brain barrier, would it not be logical to just focus on peripherally selective CB1R antagonists, since they may still act on the ARC neurons (AGRP and POMC)? So as long as the antagonist is peripherally selective, would it make a considerable difference on the neuropsychiatric outcome if its neutral/biased antagonist or an inverse agonist? Side note- I did like the way the authors went about describing these pharmacological principles in a very intuitive kind of way.
We agree with the Reviewer that peripherally restricted CB1R antagonism, at least theoretically, should be both efficacious in treating obesity/metabolic syndrome and safe at the same time. This principle is well exemplified by peripheral compound such as JD-5037 and AM6545, which have well-documented beneficial effects in the liver, pancreas, kidney and adipose tissue and are devoid of detectable behavioral effects (Tam et al., 2010, Tam et al., 2017, Jourdan et al., 2013, Jourdan et al., 2017). Indeed, these compounds likely penetrate the ARC, (Faouzi et al., Endocrinology, 2007) where they probably enhance leptin signaling directly (Szanda et al., iScience, 2023). However, conductive flow of the CSF and the glymphatic system may, in principle, distribute such primarily peripheral drugs from the ARC to remote brain areas that have a bona fide blood brain barrier (see, e.g., Hablitz et al., J.Neurosci., 2021 for review). We suppose that such a mechanism is likely not of great concern, but scientists, clinicians, pharmaceutical companies and patients will be especially wary of any CB1R antagonist `after the fall`. Therefore, adding an extra layer of safety with biased compounds, while retaining precious therapeutic potential as MRI-1891 does, may be trust inspiring towards all those groups.
- I do not quite understand why there is a different paragraph for 'phytocannabinoids'? Would it not just fall under the NAM section for CBD, or neutral antagonist section for THCV?
A: We considered this, but phytocannabinoids are a curiosity in terms of drug development – companies that have learned to extract or synthesize minor phytocannabinoids in bulk are asking ‘what can we do with them’. It’s really the polar opposite of rational drug design. Therefore, it seemed to us that they deserved their own section to highlight this and also the idea that the field has to some extent come full circle, returning to phytocannabinoids after a spell of 50 years.
- The figure 5 you have is missing a significant portion of the text because its off center. The right side of it is completely cut off resulting in missing text.
A: Sorry about that. Hopefully we have corrected the formatting in the revision.
- While I do understand that the review is on CB1R antagonists, a major shift has been made to study the CB2R instead of the CB1R since you would be circumventing the issues with CB1R-associated neuropsychiatric effects. Do you suggest that this is a good alternative or are there any drawbacks to this line of research?
A: We did have to make a decision about whether to include CB2. In view of the still ongoing debate over whether CB2 is even present in the CNS under non-pathological circumstances, we decided to leave it out. CB2 as a target will have its own raft of potential side effects since the receptor is so tightly associated with the immune system.
- Minor comment-but you have to list the full form of all abbreviations in the legend for the figures, even if its commonly employed in the molecular signaling community (Ras Raf Mek Figure 3). Also, several abbreviations in figure 2 are not listed in the legend.
A: Thank you for pointing this out. We’ve updated the text with full names of the abbreviated compounds.
- The section on potential use of rimonabant for substance abuse is a very interesting section! How does rimonabant compare in its efficacy for currently employed FDA drugs?
A: Thank you, it does seem to have been a case of throwing the baby out with the bath water though as we noted in the text (line 317) at least one approved drug, varenicline, has a substantially greater likelihood of success in helping people to stop smoking. The efficacy of many FDA approved smoking cessation therapies has been extensively reviewed (Sigaly et al., 2002, Cahill et al., 2013) we have summarized those (see table below). All appear to be superior to rimonabant when comparing the odds ratios for smoking cessation but it is possible that further research might identify a more promising compound or that a compound might prove effective as part of a combination therapy. We took caution in reporting these values in the section, as the duration of treatment for each FDA approved therapy and rimonabant (Robinson et al., 2018) varied greatly and it is difficult to draw clear conclusions on comparative efficacy. We now also briefly mention the CB1 antagonist AEF0117, which shows substantial efficacy in treating cannabis use disorder in Phase 2a trials, as a potential future drug to facilitate the treatment of cannabis intoxication. (~lines 352 and ~410)
|
Smoking Cessation Therapy |
Odds ratio (OR) for cessation of smoking after treatment a |
|
Rimonabant *1 |
OR = 1.5 |
|
Varenicline 2 |
OR = 2.88 |
|
Bupropion 2 |
OR = 1.82 |
|
Nicotine gum 3 |
OR = 1.66 |
|
Nicotine patch 3 |
OR = 1.74 |
|
Nasal spray 3 |
OR = 2.27 |
|
Nicotine lozenges 3 |
OR = 2.08 |
|
*Pooled analyses from STRATUS-US and -EU a Duration of treatment and timing with which abstinence was measured post-treatment varied between each study. 1 Robinson et al., 2018 2 Cahill et al., 2013 3 Silagy et al., 2002 |
|
- When it comes to the pathophysiological effects of CB1R activation, you do not necessarily go into much detail. You have a paragraph on the liver, and even less on the adipose tissue. Nothing much on the heart or vasculature. So I understand that it may be difficult to write about this at length, as it can disrupt the narrative theme, but you have to cite other recent reviews at least that have covered this in greater detail. This would give the readers an opportunity to have a fundamental appreciation for CB1R-mediated effects before you jump into the therapeutics side of it.
We have added several references to flesh out this section. We also included recent reviews on the role of cannabinoids and CB1Rsin cardiovascular and metabolic diseases (please see answer to Q11 also).
- Where do you see the largest market for the next generation of CB1R antagonists? Is it in obesity/weight loss still, or diabetes, fatty liver or metabolic syndrome or all of the above? Also what about cardiovascular function? Is there a scope for peripheral CB1R therapeutics in cardiovascular disease?
With the availability of efficacious and safe incretin mimetics, the potential clinical niche for CB1R antagonist has certainly become narrower in the last decade. However, many of the beneficial effects of CB1R antagonists, such as improved insulin and leptin sensitivity, increased adiponectin secretion, protective capacity against pancreatic β-cell loss, fatty liver disease and glomerulopathy (see pertinent references in the MS), are independent of body weight reduction and blood glucose normalization. This suggests that peripheral CB1R agonists may be a good candidate for combination therapy with GLP-1 analogues and/or with antidiabetics, as CB1R blockade could provide the weight reduction and glycemic control independent protection from those pathologies. We touch on this possible therapeutic niche briefly in the MS (line ~256).
The role of cannabinoids in cardiovascular disease is an important and complex topic, one that is certainly beyond the scope of our review. Instead, we included excellent recent reviews. Regarding the activation of CB1Rs, it has adverse effects in the cardiovascular system, including tachycardia and decreased contractility of the myocardium. Moreover, cannabis users have somewhat poorer clinical outcome in myocardial infarction. On the other hand, rimonabant showed early promise in the treatment of cirrhotic hypotension (Batkai et al., Nat.Med., 2001), but research into its cardiovascular actions also suffered a blow once it was withdrawn from the market. (However, recent work of Hunyady and Szekeres on the interplay between the renin-angiotensin-aldosterone system and vascular endocannabinoids may be of interest to the Reviewer and cardiovascular researchers.). It was, nonetheless, well-documented in human trials that the favorable metabolic effects of rimonabant translate into substantially reduced cardiovascular risk (Van Gaal et al., Lancet, 2005).
- It is interesting that you do not describe the full signaling cascade of CB1R in the text, but do so in the figure only (figure 3). And this is not just in neurons, CB1R-mediated inhibition of cAMP generation is observed in virtually all cells where the receptor is expressed. Also several non-canonical signaling pathways exist for CB1R. Since you also talk about Barr2 in a later section, it becomes slightly important to actually go into detail and explain both the canonical (Gi coupled) and non-canonical signaling pathways (Barr1 and 2, and other G proteins) of CB1R.
We have added detail about the signaling pathways to the text (lines ~100-103) and a bit more information to the legend for figure 3.
